

# Data augmentation and deep neural networks for the classification of Pakistani racial speakers recognition

Ammar Amjad[1], Lal Khan[1] and Hsien-Tsung Chang[1,2,3,4]

[1] Department of Computer Science and Information Engineering, Chang Gung University, Taoyuan, Taiwan
[2] Bachelor Program in Artificial Intelligence, Chang Gung University, Taoyaun, Taiwan
[3] Department of Physical Medicine and Rehabilitation, Chang Gung Memorial Hospital, Taoyuan, Taiwan
[4] Artificial Intelligence Research Center, Chang Gung University, Taoyuan, Taiwan

Corresponding author
Hsien-Tsung Chang,
smallpig@widelab.org

## ABSTRACT

Speech emotion recognition (SER) systems have evolved into an important method for recognizing a person in several applications, including e-commerce, everyday interactions, law enforcement, and forensics. The SER system's efficiency depends on the length of the audio samples used for testing and training. However, the different suggested models successfully obtained relatively high accuracy in this study. Moreover, the degree of SER efficiency is not yet optimum due to the limited database, resulting in overfitting and skewing samples. Therefore, the proposed approach presents a data augmentation method that shifts the pitch, uses multiple window sizes, stretches the time, and adds white noise to the original audio. In addition, a deep model is further evaluated to generate a new paradigm for SER. The data augmentation approach increased the limited amount of data from the Pakistani racial speaker speech dataset in the proposed system. The seven-layer framework was employed to provide the most optimal performance in terms of accuracy compared to other multilayer approaches. The seven-layer method is used in existing works to achieve a very high level of accuracy. The suggested system achieved 97.32% accuracy with a 0.032% loss in the 75%:25% splitting ratio. In addition, more than 500 augmentation data samples were added. Therefore, the proposed approach results show that deep neural networks with data augmentation can enhance the SER performance on the Pakistani racial speech dataset.

## INTRODUCTION

Speaker emotion recognition (SER) is an attractive study since there are still many issues to address and many research gaps that need to be filled. However, deep learning (DL) and machine learning (ML) approaches have tackled SER challenges, particularly in research that employs speech datasets with enormous volumes of data. The amount of data is increasing by the moment. Consequently, an expansion in the amount of data worldwide is inevitable. Social websites, personal archives, sensors, mobile devices, cameras, webcams, financial market data, and health data create hundreds of petabytes of data (*Gupta & Rani, 2019*; *Khan et al., 2022a*). By 2025, the World Economic Forum predicts that the world will

create 463 exabytes of data every day. Finding the appropriate method to convert such a large volume of data into useful information is difficult.

Therefore, artificial intelligence (AI) has been used in numerous fields of the latest studies. Previously, speech recognition studies utilizing ML achieved a high degree of precision by using the Gaussian mixture model (GMM) technique (*Marufo da Silva, Evin & Verrastro, 2016*; *Maghsoodi et al., 2019*; *Mouaz, Abderrahim & Abdelmajid, 2019*), and the hidden Markov model (HMM) technique (*Veena & Mathew, 2015*; *Bao & Shen, 2016*; *Chakroun et al., 2016*; *Maurya, Kumar & Agarwal, 2018*). However, as the data increases, the level of accuracy with these techniques drops rapidly, to the point where these traditional ML approaches suffer from low accuracy and generalization issues (*Xie et al., 2018*). Nevertheless, this technique provides a reliable strategy for addressing data groupings, making it appropriate for various situations.

Several studies have been conducted regarding SER based on deep learning using different methods, such as the deep neural network (DNN) (*Seki, Yamamoto & Nakagawa, 2015*; *Najafian et al., 2016*; *Matjka et al., 2016*; *Dumpala & Kopparapu, 2017*; *Snyder et al., 2018*; *Najafian & Russell, 2020*; *Rohdin et al., 2020*; *Khan et al., 2021*; *Amjad, Khan & Chang, 2021b*, *2021a*; *Khan et al., 2022b*) and convolutional neural network (CNN) methodologies used in the study (*Ravanelli & Bengio, 2019*) attained an overall accuracy of 85% with the TIMIT database and 96% with LibriSpeech. Using the deep learning technique, *An, Thanh & Liu (2019)* obtained 96.5 percent accuracy and significantly improved the ability to handle multiple issues in SER. However, DL requires a lot of training datasets, which are challenging to gather and expensive. Therefore, this approach is unsuitable for SER utilization because it will yield overfitting problems and may lead to skewed data. The use of data augmentation (DA) is one solution to the problem of small data in the SER study. A DA approach is a technique that can be used to create additional training datasets by altering the shape of a training dataset. DA is helpful in many investigations, such as digital signal processing, object identification, and image classification (*Wu, Chang & Amjad, 2020*; *Li et al., 2020*; *Amjad et al., 2022*).

The DA technique has been extensively used in various fields of study because a few samples in many different DA classes can help solve a problem more effectively (*Zheng, Ke & Wang, 2020*). For example, multiple SER studies using DA (*Schlüter & Grill, 2015*; *Salamon & Bello, 2017*; *Pandeya & Lee, 2018*) showed a reduction of up to 30% in classification errors and obtained 86.194% accuracy. Data augmentation includes several approaches that have been effectively used in various research, including generative adversarial networks (GANs) and variational autoencoders (VAEs) approaches (*Moreno-Barea, Jerez & Franco, 2020*). The suggested approach obtained accuracy using limited data, with 87.7 percent. In another investigation, scientists employed an auditory DA strategy to achieve an 82.6 percent accuracy for Mandarin-English code flipping (*Long et al., 2020*). As presented in *Ye et al. (2020)* pitch shifting is frequently utilized in DA and achieved 90% accuracy. In addition, *Damskägg & Välimäki (2017)* employed the time-stretched data augmentation approach when performing DA-based fuzzy identification on various audio signals. *Aguiar, Costa & Silla (2018)* incorporated Latin music's noise usage, shifting the pitch, loudness variation, and stretching the time to further enhance genre

categorization. As a result, *Rituerto-Gonzlez et al. (2019)* reported an 89.45 percent accuracy using the database (LMD). We propose DA because it is proven to increase the quantity of the dataset so that it can help improve speaker recognition performance with an accuracy rate of 99.76.

The proposed study presents a data augmentation method based on a seven-layer DNN for recognizing racial speakers in Pakistan by utilizing 400 audio samples from multiple classes of racial speakers in Pakistan. However, this kind of study may easily lead to multiclass difficulties due to the many classes it includes. On the other hand, DNN approaches are often utilized in SER (*Nassif et al., 2019*). In addition, DNN is also a powerful model capable of achieving excellent performance in pattern recognition (*Nurhaida et al., 2020*). The study was undertaken by *Novotny et al. (2018)* in conjunction with Mel-frequency cepstral coefficients (MFCC) has shown the effectiveness of DNN in SER and improved network efficiency in busy and echo conditions. Furthermore, DNN with Mel-frequency cepstral coefficients has outperformed numerous other research approaches on SER single networks (*Saleem & Irfan Khattak, 2020*). Additionally, DNN has been effectively fusing with augmented datasets. The presented approach employs a seven-layer neural network because the seven-layer technique yields the highest efficiency and accuracy when used in previous works with an average precision above 90% (*Liu, Fang & Wu, 2016*; *Zhang et al., 2018*; *Li et al., 2019*). Furthermore, including the Pakistani speakers with many classes employing DNN with DA would improve the identification efficiency of multiple emotional classes.

This article is divided into sections. The Introduction describes the significant issue and the studies done by the speaker; 'Related works' includes many existing works that support the proposed study; 'Data augmentation' describes data augmentation and several methodologies that are used in the research. The next section discusses DNNs, and the deep learning techniques employed. The methodology is covered in the next section, followed by the research outcomes and a discussion. Finally, the 'Conclusion' section covers various significant things about the conclusion of the research outcomes.

## RELATED WORKS

The proposed study on multi-racial voice recognition was carried out in many nations, like China (*Nassif et al., 2019*), Africa (*Oyo & Kalema, 2014*), Italy (*Najafian & Russell, 2020*), Pakistan (*Syed et al., 2020*; *Qasim et al., 2016*), the United States (*Upadhyay & Lui, 2018*), and India, through CNN and MFCC (*Ashar, Bhatti & Mushtaq, 2020*). It is a vital technique that many researchers have chosen to enhance SER efficacy (*Chowdhury & Ross, 2020*).

In contrast, the limitations of multi-racial SER systems investigated in some studies included limited speech data and a lack of emotional classes. Therefore, weak data training methods may result from inaccurate outcomes. Nevertheless, some research in SER and multi-racial SER systems, such as automatic Urdu speech recognition using HMM, involves a 10-speaker category consisting of eight male and two female speakers with 78.2 percent accuracy. In addition, the study of multilingual, multi-speaker involves three classes, namely Javanese, Indonesian, and Sundanese (*Azizah, Adriani & Jatmiko, 2020*).

However, this investigation has limits regarding the number of emotional categories. Various types of SER studies have been conducted. For example, *Durrani & Arshad (2021)* used deep residual network (DRN) with a 74.7 percent accuracy rate. Another study employing MFCC and Fuzzy Vector Quantization Modeling on hundred categories from the TIMIT database gives 98% accuracy, higher than other approaches such as Fuzzy Vector Quantization two and Fuzzy C-Means (*Singh, 2018*). The ML technique is still utilized in conjunction. The classic approaches, such as the HMM, recognize four Moroccan dialect speakers using 20 speakers; this research achieved a 90% accuracy rate for speaker recognition (*Mouaz, Abderrahim & Abdelmajid, 2019*).

A single-layer DNN with a data augmentation approach was also utilized to investigate the impact of stress on the performance of SER systems, obtaining an accuracy of 99.46% with the VOCE database (*Rituerto-Gonzlez et al., 2019*). The VOCE database comprises 135 utterances from forty-five speakers. In addition, the GMM and MFCC with the TIMIT database were utilized to recognize short utterances from 64 different regions and obtained 98.44% accuracy (*Chakroun & Frikha, 2020*). This accuracy is higher than the traditional GMM. Another approach was employed in a study (*Hanifa, Isa & Mohamad, 2020*) that used 52 recordings of Malaysian recorded samples utilizing the MFCC in the feature extraction, with an accuracy of 57%. Along with machine learning, numerous works in SER and multi-racial utilize the DL technique, regarded as a rigorous approach to SER. The Deep Learning technique with a deep neural network is used with different techniques, one of which is DA, as demonstrated in a study presented by *Long et al. (2020)* on the OC16-CE80 dataset. This Mandarin-English mixlingual speech *corpus* successfully produced an effective model for SER with an 86% accuracy. The above research has several similarities with the proposed study: the dataset containing speakers from multi-racial backgrounds, DA, and the MFCC feature extraction method. However, some preceding studies differed from the proposed study in many ways, including the number of speech categories, the length of the utterance, and the identification techniques utilized. Table 1 explains the evolution of work on SER in further detail.

## DATA AUGMENTATION

Researchers employ a method known as data augmentation to enhance the number of dataset samples. DA is an approach for increasing the number of training datasets useful for neural network training (*Rebai et al., 2017*) and has a major influence on deep learning with limited datasets (*Ma, Tao & Tang, 2019*). Furthermore, DA is a useful method for overcoming overfitting problems, enhancing model dependability, and increasing generalization (*Wang, Kim & Lee, 2019*), which are common issues in machine learning. Research based on deep learning with data augmentation techniques is critical for improving prediction accuracy while dealing with massive volumes of data (*Moreno-Barea, Jerez & Franco, 2020*). There are a few data augmentation methods, including adding white noise into an original sample, shifting the pitch, loudness variation, multiple window sizes, and stretching the time. The small size of the dataset is a problem when utilizing deep learning approaches. The proposed approach used to overcome this issue is to induce noise into the training data.

**Table 1 Detailed description of datasets.**

| Reference | Approach | Database | Classes | Accuracy |
|---|---|---|---|---|
| *Wang, Wang & Liu (2014)* | HMM and GMM | S-PTH database | 4 | 13.8% and 24.6% error rate |
| *Najafian et al. (2016)* | DNNs | The First Accents of the British Isles Speech *Corpus* | 14 | 3.91% and 10.5% error rate |
| *Qasim et al. (2016)* | Support Vector Machine,Random Forest and Gaussian Mixture Model | Recorded Pakistan ethnic speaker | 6 | 92.55% |
| *Salamon & Bello (2017)* | SB-CNN | Urban- Sound8K | 10 | 94% |
| *Upadhyay & Lui (2018)* | Deep Belief Network | FAS Database | 6 | 90.2% |
| *Singh (2018)* | Fuzzy Vector Quantization | TIMIT | 100 | 98.8% |
| *Mouaz, Abderrahim & Abdelmajid (2019)* | HMM One layer Deep Neural Network | VOCE *Corpus* Dataset | 4 | 90% |
| *Ashar, Bhatti & Mushtaq (2020)* | CNN | Spontaneous Urdu dataset | – | 87.5% |
| *Azizah, Adriani & Jatmiko (2020)* | DNNs | Indonesian speech *corpus* | 4 | 98.96% |
| *Chakroun et al. (2016)* | GMM | TIMIT | 8 | 98.44% |
| *Hanifa, Isa & Mohamad (2020)* | Support Vector Machine | speaker ethnicity | 4 | 56.96% |
| *Hanifa, Isa & Mohamad (2020)* | DNN | OC16 | 2 | 86.10% |

**Adding white noise:** Adding white noise to a speaker's data enhancements recognition effectively (*Ko et al., 2017*). This approach involves the addition of random sound samples with similar amplitude but various frequencies (*Mohammed et al., 2020*). Using white noise in a speech signal increases the performance of SER (*Schlüter & Grill, 2015*; *Aguiar, Costa & Silla, 2018*; *Hu, Tan & Qian, 2018*). Furthermore, when white noise is added to an original sound gives a distinct sound effect, which increases the performance of SER.

**Pitch shifting:** is a commonly used method in an audio sample to increase or decrease the original tone of voice. Pitch variations are performed using this technique without affecting playback speed (*Mousa, 2010*). In addition, a method is utilized in pitch shifting to increase the pitch of the original sound without changing the duration of the recorded sound clip (*Rai & Barkana, 2019*). For example, various studies on singing voice detection (SVD) (*Gui et al., 2021*), environmental sound classification (ESC) (*Salamon & Bello, 2017*), and domestic cat classification have shown that pitch shifting may be highly effective for DA (*Pandeya & Lee, 2018*).

**Time stretching:** is a way to change the speed or length of an audio signal without changing the tone. Instead, it is used to manipulate audio signals (*Damskägg & Välimäki, 2017*). This technique is suitable for analyzing auditory signals that comprise tone, noise, and temporal elements. Numerous investigations used time stretching with other approaches such as synchronous overlap, fuzzy, and CNN to increase the efficiency of the suggested framework (*Sasaki et al., 2010*; *Kupryjanow & Czyżewski, 2011*; *Salamon &*

*Bello, 2017*). These studies used different techniques, such as the synchronous overlap algorithm, fuzzy logic, and CNN, to improve the performance of the proposed model.

**Multiple window size:** Multiple window size features are retrieved from a windowed signal called frames. The window strongly influences the obtained features retrieved from the voice signal-based functions width since signals are often steady for limited periods (*Kelly & Gobl, 2011*). Suppose the length of the window is relatively small. In that case, insufficient training datasets are available to get an accurate spectrum for estimating the signals. On the other hand, if the window's length is set very wide, the signal may vary significantly across the frame. Thus, determining the width of the window function is a critical phase that is made more difficult by the lack of details about the original data (*Rabiner & Schafer, 2007*; *Zhang et al., 2019*). Several studies have demonstrated that the optimal window size selection contributes to the correlation between the acoustic representation and the human perception of a speech signal (*Nisar, Khan & Tariq, 2016*; *Kirkpatrick, O'Brien & Scaife, 2006*). Three tuples express a window function: width of the window, offset, and shape. To extract a part of a signal, multiply the signal's value at the time "t," signal[t], by the value of the hamming window at a time "t," window[t], which is expressed as: windowsignal[t] = window[t] * signal[t].

A windowed signal is utilized to create characteristics for emotion recognition. For SER, a standard size window of 25 ms is employed to extract features with a 10 ms overlap (*Yoon, Byun & Jung, 2018*; *Tarantino, Garner & Lazaridis, 2019*; *Ramet et al., 2018*). On the other hand, some research has indicated that a larger window size improves emotion identification performance (*Chernykh & Prikhodko, 2018*; *Tripathi, Tripathi & Beigi, 2019*). In addition, other studies have assessed the significance of step size (overlap window size). However, SER analysis is conducted using a single-window (*Tarantino, Garner & Lazaridis, 2019*; *Chernykh & Prikhodko, 2018*). *Tarantino, Garner & Lazaridis (2019)* investigated the influence of overlap window size on SER. They discovered that a small step size leads to a lower test loss. *Chernykh & Prikhodko (2018)*, explored multiple window widths ranging from 30 to 200 ms before settling on a unique 200 ms window for the SER study.

## METHODOLOGY

Deep Learning has been used to create a variety of solid approaches for SER. The DNN is one of the most widely utilized deep learning approaches. In many SER studies, deep neural networks are employed because they have several benefits over conventional machine learning approaches. There are several benefits to using the DNN approach in many scientific domains, including object detection, geographic information retrieval, and voice classification (*Seifert et al., 2017*). The DNN-based acoustic model was used in previous work to achieve high-level performance (*Seki, Yamamoto & Nakagawa, 2015*; *Snyder et al., 2018*; *Novotny et al., 2018*; *Saleem & Irfan Khattak, 2020*).

The structure of a DNN approach is composed of input, hidden, dropout, and output layers (*Rajyaguru, Vithalani & Thanki, 2020*). The deep neural network is an evolution of the neural network (see Fig. 1), which is essentially a function in a mathematical measure R: A ⇒ B that may be stated as follows.
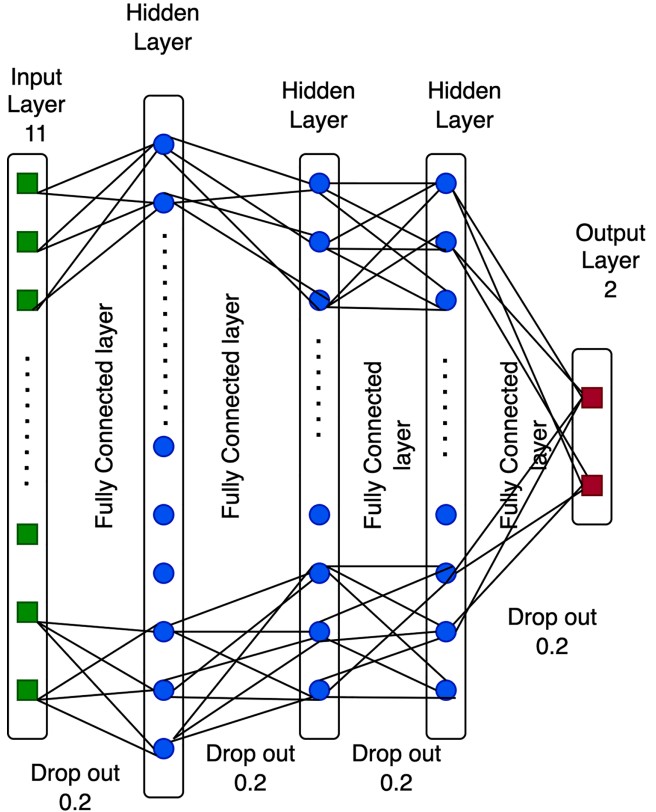

**Figure 1** **Structure of a deep neural network.**

### Input layer

The input layer comprises nodes that obtain the inputted data from variable A. These nodes are directly connected to the hidden units. The generation of eleven input layer features is generated after a preprocessing step utilizing the principal component analysis (PCA) algorithm.

### Hidden layer

The hidden layer is composed of nodes that obtain data from the first layer. Previous studies have suggested that the volume of nodes in the hidden layer may be influenced by the dimensions of the input and output layers. For example, in Fig. 1, the size of the hidden neurons is 24.12, and 12 in the hidden units, which is the optimal number of deep neural network characteristics based on previous studies.

### Dropout (DO)

A dropout is a single approach utilized to generate a range of system designs that may be used to address overfitting issues in the model. The dropout value ranges between 0 and 1. Dropout is set to a size of 0.2 for each layer in Fig. 1, since DNN obtains the highest efficiency with this value.

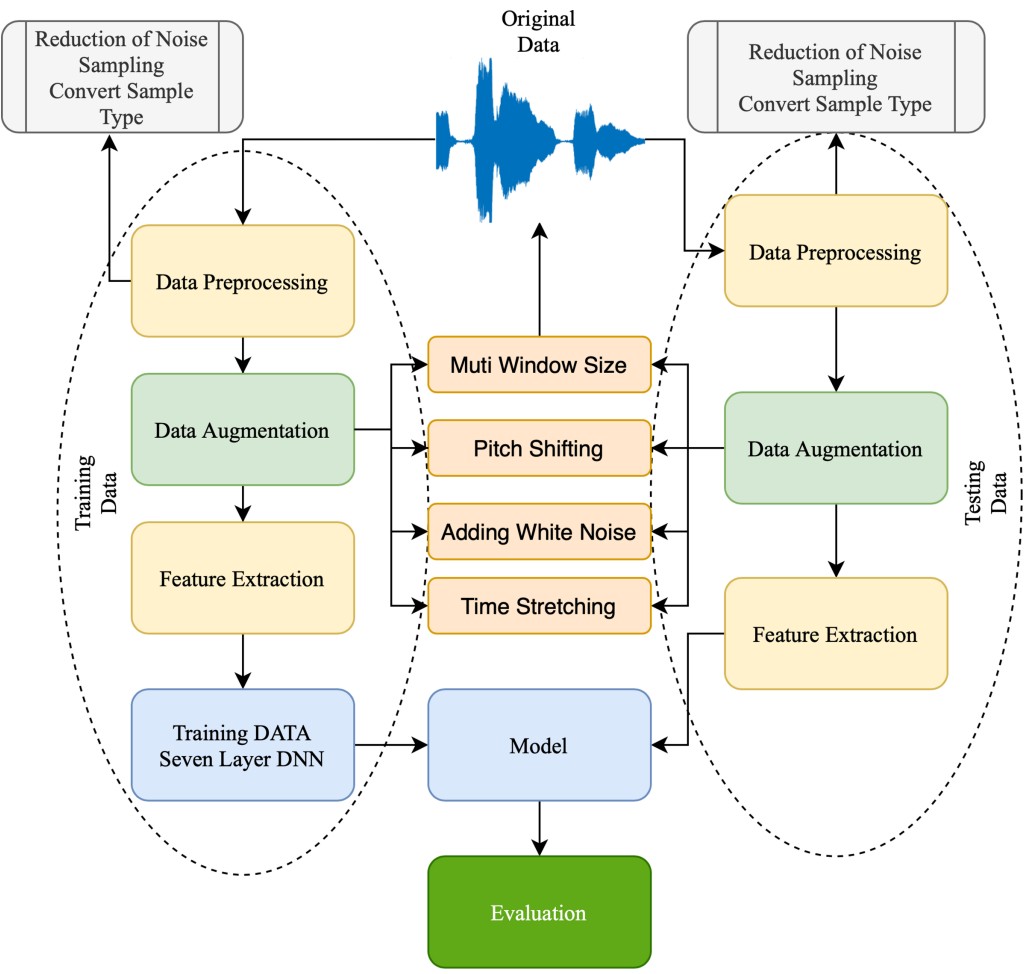

**Figure 2** **Structure of proposed approach.**

## Output layer

The output layer comprises nodes that access data directly from the hidden or input layer. The output value provides a computation outcome from the A to B value. For example, the two output layer nodes in 1 represent the number of groups. The proposed technique improved the Pakistani racial speaker recognition accuracy. It was based on the seven-layer DNN architecture with a data augmentation approach. Figure 2 illustrates the proposed method's architecture. The proposed SER using a seven-layer DNN-DA approach to the multi-language dataset, as shown in Fig. 2, is a robust approach. First, a dataset is divided into training data (75% of the dataset) and testing data (25% of the dataset). Then, the training data is preprocessed by trimming audio signals with identical temporal lengths and generating sample types with similar shapes and sizes. Moreover, four techniques of the data augmentation procedure are performed on the dataset to enhance audio data. Finally, the MFCC extracts and processes the features with a seven-layer DNN-DA for classification. The testing dataset performs the same preprocessing steps, data augmentation, and feature extraction using MFCC. Furthermore, the proposed approach will be evaluated using testing data to see how accurate it is speaker recognition.

**Table 2 Duration of audio speech data in hours.**

| Racial | Number of male and female speakers | Duration per sample | Number of samples | Nature of samples |
|---|---|---|---|---|
| Punjabi (*Wang & Guan, 2008*) | 4 males and 4 females | 42 s | 500 samples | Speaker and text independent |
| Urdu (*Wang & Guan, 2008*; *Syed et al., 2020*) | 4 males and 4 females | 42 s | 500 samples | Speaker and text independent |
| Sindhi (*Syed et al., 2020*) | 32 males and 38 females | 30 s | 80 samples | Speaker and text independent |
| Saraiki | 42 males and 28 females | 30 s | 80 samples | Speaker and text independent |
| Pashto | 35 males and 35 females | 30 s | 80 samples | Speaker and text independent |

## DATASET AND PREPROCESSING

This study utilized a dataset of Pakistan's five most spoken local languages. The information was obtained to adjust for the numerous ethnicities. Various online resources were used to compile this dataset (*Wang & Guan, 2008*; *Syed et al., 2020*). This study aims to gather data from areas of Pakistan where Urdu and its five primary ethnicities (Punjabi, Sindhi, Urdu, Saraiki, and Pashto) are spoken. The audio samples were processed using PRAAT software. The dataset for the Urdu language is summarized in Table 2. The dataset is utilized only to recognize Urdu racials. The dataset contains 80 distinct utterances for each ethnicity type with different levels of education, ranging from semi-literate to literate. Each audio file is from an individual speaker, resulting in 80 distinct speakers per ethnic group. Each clip is 30 s long, in mono channel WAV format, and sampled at 16 kHz of Sindhi, Saraiki, and Pashto languages. Additionally, each utterance is distinct from others in the dataset. The dataset includes sounds from 80 speakers of five racials, for 1,240 clips.

The dataset processing uses a segmentation process similar to that used for the dataset of the Ryerson Audio-Visual Database of Emotional Speech and Song (RAVDESS). This multimodal recording dataset takes the form of emotional speech and songs recorded in audio and video formats (*Atmaja & Akagi, 2020*). Experiments on RAVDESS were carried out by *Livingstone & Russo (2018)*, and they involved the participation of 24 professional actors with North American accents. The research included speech and songs with various facial expressions, including neutral, calm, happy, sad, angry, fearful, surprised, and disgusted. In the data of Pakistani racial speakers, the complete audio utterances are segmented once again using the approach that is described below:

- Modality 001 = only-audio, 002 = only-video, 003 = audio-video
- Classes: 001 = disgust, 002 = neutral, 003 = fearful, 004 = angry, 005 = happy, 006 = surprised, 007 = sad, 008 = calm
- Vocal: 001 = song, 002 = speech
- Intensity: 001 = strong, 002 = normal
- The racial of the speakers as a class from 01 to 5
- Repetition: 001 = First, 002 = second
- Speaker sequence number per tribe/region from 01 to 10

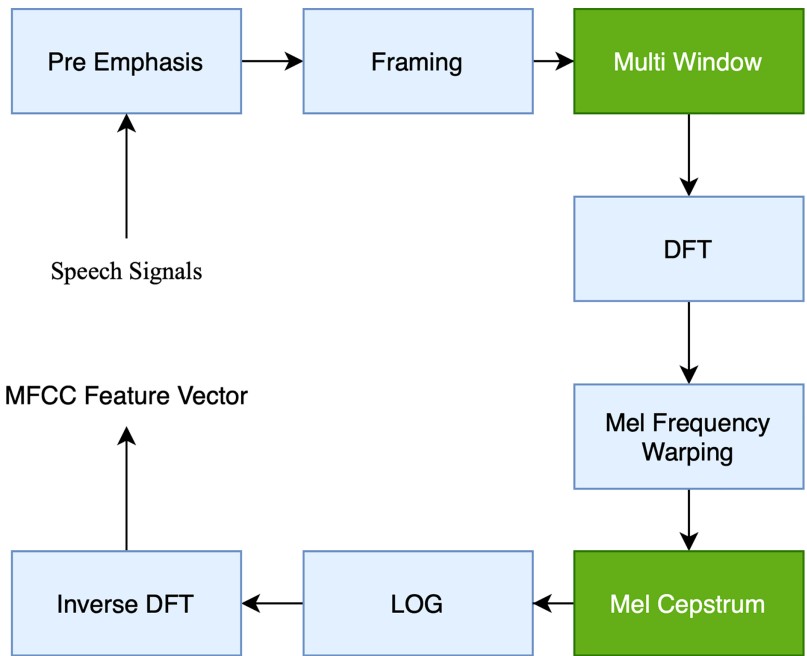

**Figure 3  Block diagram of the computation steps of MFCC.**

## Feature extraction

We employed MFCC in the proposed study since it is one of the most robust approaches to extracting features from SER features. MFCC is the most widely used approach for obtaining spectral information from a speech by processing the Fourier Transform (FT) signal with a perception-based Mel-space filter bank. Additionally, in the proposed study, Librosa is used to extract MFCC features. This Python library has functionality for reading sound data and assisting in the MFCC feature extraction method. According to *Hamidi et al. (2020)*, the MFCC technique is shown in Fig. 3: The MFCC approach enhances the audio sound input during the preemphasis phase and increases the signal-to-noise ratio (SNR) enough to ensure that the voice is not influenced by noise. The framing mechanism divides the audio signal into many frames with the same signal count. Windowing is the technique of employing the window function to weight the output frame. The following procedure is the DFT (discrete Fourier transform), which examines the frequency signal derived from the discrete-time signal. Then, the MFCC obtained from the original utterances is determined using the filter bank (FB). The wrapping of Mel Frequency is often used in conjunction with a FB. A FB is a kind of filter used to determine the amount of energy contained within a certain frequency range, *Afrillia et al. (2017)*. Finally, the logarithmic (LOG) value is obtained by converting the DFT result to a single value. Inverse DFT is a technique for obtaining a perceptual autocorrelation sequence based on the linear prediction (LP) coefficient computation. The MFCC technique was employed in this study by setting frame lengths at 25 with a hamming window, 13 spectral and 22 lifter coefficients, and 10 frameshifts. The MFCC approach enhances the audio sound input during the preemphasis phase, increasing the signal-to-noise ratio (SNR) enough to ensure

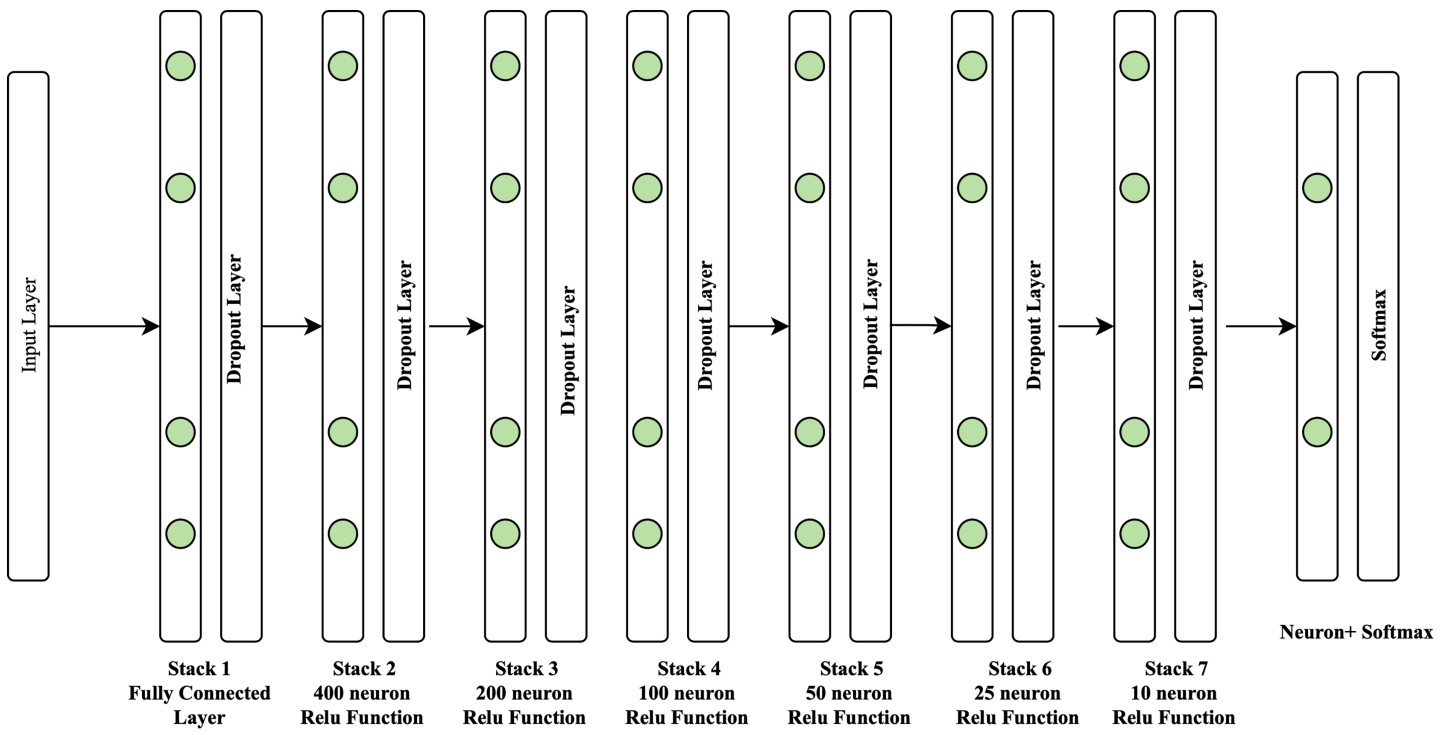

**Figure 4 Structure of proposed approach.**

that the voice is not influenced by noise. The framing mechanism divides the audio signal into many frames with the same signal count. Windowing is the technique of employing the window function to weigh the output frame. The following procedure is the DFT (discrete Fourier transform), which examines the frequency signal derived from the discrete-time signal. Then, the MFCC obtained from the original utterances is determined using the filter bank (FB). The wrapping of Mel Frequency is often used in conjunction with a FB.

### Seven layer DNN

In this study, the rectified linear unit (Relu) activation function is utilized in conjunction with the Adam optimizer (AO). Adam optimizer is used to improve the learning speed of deep neural networks. This algorithm was introduced at a renowned conference by deep learning experts *Kingma & Ba (2017)*, with a 0.2% dropout rate. A deep neural network comprises seven layers, with the structure shown in Fig. 4.

As seen in Fig. 4, the seven-layer architecture of the DNN consists of one fully connected layer with 400 neurons on layer two, which is the expected volume of neurons identified in our investigation. The following layer has just half of the neurons from the preceding layer. Layer one is composed of dense functions that create a fully connected layer. The second layer comprises 400 neurons composed of the dense and dropout functions used in the neural network to avoid overfitting and accelerate the learning process. The third layer comprises 200 neurons. The fourth layer comprises 100 neurons,

the fifth layer comprises 50 neurons, and the sixth layer comprises 25 neurons. It is also composed of dense and dropout functions. Finally, the seventh layer comprises 10 neurons with dense and dropout functions. At the same time, softmax activation is used as the output layer. The seven-layer DNN architecture is employed in this work because it provides the maximum level of accuracy compared to the three-layer DNN and five-layer DNN.

### Evaluation

Acted, semi-natural, and spontaneous datasets were employed in the proposed study. In addition, the split ratio method with train test split assessment was used to evaluate performance in ML. The proposed approach separates the data into training for matching the ML architecture and testing the ML architecture. The most utilized ratio is splitting training and testing data by 70%:30%, 80%:20%, or 90%:10%. Multiple factors determine the split ratios, namely the compute costs associated with the model training, the computational costs associated with testing the model, and data analysis. Accuracy is a commonly used metric for assessing the extent of incorrectly identified items in balanced and approximately balanced datasets (*Atmaja & Akagi, 2020*). It is one of the model performance assessment methodologies often used in ML.

## RESULTS AND DISCUSSION

This study utilized DA methods to evaluate a Pakistani racial speech dataset using a 44,100 mono sample rate. The testing efficacy of the seven-layer DNN-DA approach at epoch 100 with batch size two is illustrated in Fig. 5. Testing a training dataset yields an accuracy of 97.32% with a total loss of 0.03. As shown in Fig. 5, the total loss decreases from epoch 1 to 100. However, it has remained unstable at epochs 20, 28, 38, 64, 73, and 77, with loss increases that automatically decrease precision efficiency at epochs 20, 28, 38, 64, 73, and 77. It eventually stabilized above 90% in the 88th epoch. The graph in Fig. 6 illustrates the outcomes of model testing utilizing data testing. Using 500 data wav files shows that the seven-layer DNN-DA model produces a robust technique for SER. With highest efficiency of 97.32% and a low loss rate of 0.032, the seven-layer DNN-DA model produces a robust approach for speaker recognition and lacks overfitting in this model test. A split ratio is also used to assess the proposed approach performance, as illustrated in Table 3.

According to Table 4, when the split ratio is 75:25, the trained model achieves the highest accuracy and the lowest loss level. As shown in Table 5, the accuracy of the results decreases when the split ratio is 80:20. At the same time, the loss increases. Finally, when the split ratio is 90:10, the accuracy results increase while the loss rate decreases. Table 6 results illustrate that testing with a large amount of training data is beneficial since it exposes the model to many instances, allowing it to identify the optimal solution.

However, if we utilize an insufficient training dataset, the model will lack expertise, resulting in inferior output during testing. The proposed approach will gain a more profound understanding and increase the model's generalizability by including many

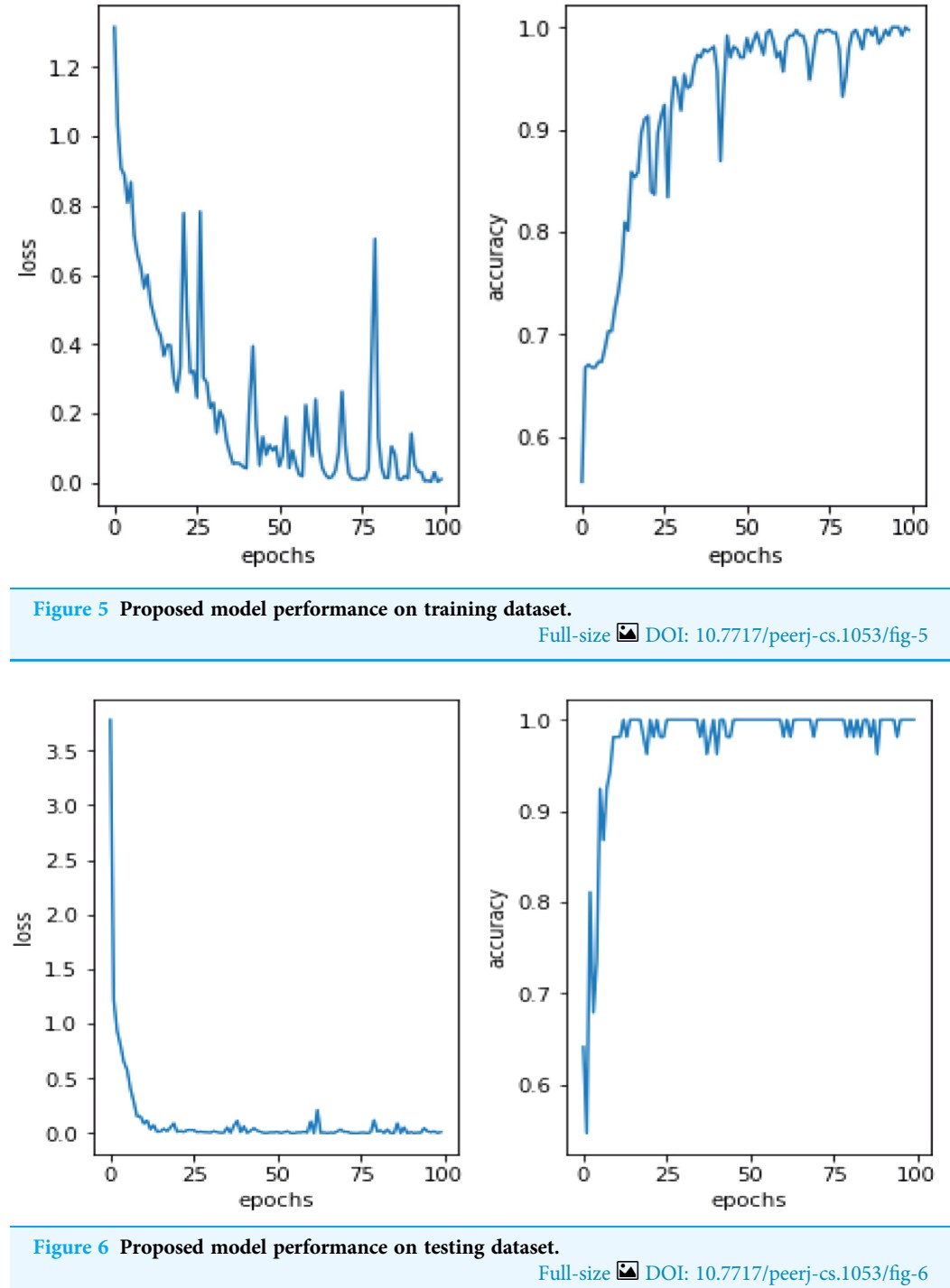

**Figure 5 Proposed model performance on training dataset.**

**Figure 6 Proposed model performance on testing dataset.**

testing datasets. As shown in Tables 4–6, another test was conducted by adding 100 to 500 data samples to the original 400 wav data using the split ratio approach.

In the suggested method, a dataset with a data augmentation of 500 samples and a split ratio of 75:25 obtained the highest performance with a low total loss. However, as the sample of DA decreases, the SER model's performance decreases. In another comparison,

**Table 3 Comparison table of loss at dividing ratio with accuracy.**

| Dividing ratio | Classification accuracy | Total loss |
|---|---|---|
| 90:10 | 93.55 | 0.105 |
| 80:20 | 95.767 | 0.093 |
| 75:25 | 97.32 | 0.032 |

**Table 4 The accuracy and loss comparison table includes augmentation data with 75:25 ratio.**

| Data augmentation | Accuracy | Loss |
|---|---|---|
| 100 | 96.57 | 1.33 |
| 200 | 96.21 | 0.05 |
| 300 | 96.83 | 2.77 |
| 400 | 96.45 | 0.035 |
| 500 | 97.32 | 0.031 |

**Table 5 The accuracy and loss comparison table includes augmentation data with 80:20 ratio.**

| Data augmentation | Accuracy | Loss |
|---|---|---|
| 100 | 95.12 | 6.33 |
| 200 | 95.99 | 0.04 |
| 300 | 96.13 | 0.19 |
| 400 | 96.29 | 0.66 |
| 500 | 97.09 | 2.77 |

**Table 6 The accuracy and loss comparison table includes augmentation data with 90:10 ratio.**

| Data augmentation | Accuracy | Loss |
|---|---|---|
| 100 | 95.21 | 0.13 |
| 200 | 96.90 | 0.28 |
| 300 | 96.34 | 3.22 |
| 400 | 96.99 | 6.23 |
| 500 | 97.01 | 5.232 |

accuracy improves when a large DA and a significant amount of training data are used. Additionally, as seen in Table 7, the study has the highest accuracy performance compared to numerous methodologies using ML and DL algorithms. The study performance on SER in Table 7 demonstrates that the seven-layer approach we presented is practical. DNN-DA is a robust approach for usage in SER that has achieved a high degree of accuracy. It is not straightforward to get accurate prediction findings while researching several classes. Certain aspects of multi-classes will be more challenging since they must discriminate

**Table 7 Comparison of outcomes with different ML and DL algorithms.**

| Dataset | Classification accuracy | Accuracy |
|---|---|---|
| Pakistani racial speaker classification | KNN | 81.99 |
| | Random Forest | 71.56 |
| | Multilayer Perceptron (MLP) | 91.45 |
| | Decision Tree | 67.45 |
| | Three layers Deep Neural Network | 92.56 |
| | Five layers Deep Neural Network | 94.78 |
| | Seven Layer DNN-DA (Proposed) | 97.732 |

between many classes while generating predictions (*Silva-Palacios, Ferri & Ramírez-Quintana, 2017*). However, seven layer DNN-DA outperforms conventional machine learning methods such as k-nearest neighbors (KNN), random forest (RF), multilayer perceptron, decision tree, and DL approaches using three-layer DNN layer and five-layer DNN, as demonstrated by the highest accuracy performance compared to other approaches using three-layer DNN and five-layer layers DNN layer.

## CONCLUSION

A study in SER that includes significant data is a challenging research issue; the Pakistani racial speech dataset is comprised of utterance groups. Therefore, seven-layer DNN-DA is the approach presented in this report, which combines the data augmentation technique with a DNN to improve performance and minimize overfitting issues. Finally, some of the contributions to our work include using a Pakistani racial speech dataset in this study. Furthermore, DA can increase the amount of data by using white noise, variable window widths, pitch-shifting, and temporal stretching methods to generate new audio data for the segments. Furthermore, classification with deep neural networks of seven layers is beneficial for improving the performance of the SER system when used with all Pakistani racial speech datasets. In addition, the proposed model with the seven-layer DNN-DA technique also has an accuracy advantage, similar to some approaches using conventional ML and DL methods that also produce high accuracy performance.

### Funding
The authors received no funding for this work.

### Competing Interests
The authors declare that they have no competing interests.

### Author Contributions
- Ammar Amjad conceived and designed the experiments, performed the experiments, analyzed the data, performed the computation work, prepared figures and/or tables, authored or reviewed drafts of the article, and approved the final draft.

- Lal Khan conceived and designed the experiments, authored or reviewed drafts of the article, and approved the final draft.
- Hsien-Tsung Chang conceived and designed the experiments, performed the experiments, analyzed the data, prepared figures and/or tables, authored or reviewed drafts of the article, and approved the final draft.

### Data Availability

The code is available in the Supplemental File.

The third-party datasets are available at:

- http://shachi.org/resources/4965

- Zafi Sherhan Syed, Sajjad Ali Memon, Muhammad Shehram Shah and Abbas Shah Syed, "Introducing the Urdu-Sindhi Speech Emotion *Corpus*: A Novel Dataset of Speech Recordings for Emotion Recognition for Two Low-Resource Languages" International Journal of Advanced Computer Science and Applications (IJACSA), 11(4), 2020. DOI 10.14569/IJACSA.2020.01104104.

- Z. Xie and L. Guan, "Multimodal Information Fusion of Audio Emotion Recognition Based on Kernel Entropy Component Analysis," 2012 IEEE International Symposium on Multimedia, 2012, pp. 1-8, DOI 10.1109/ISM.2012.9.

- Zhibing Xie and L. Guan, "Multimodal information fusion of audiovisual emotion recognition using novel information theoretic tools," 2013 IEEE International Conference on Multimedia and Expo (ICME), 2013, pp. 1-6, DOI 10.1109/ICME.2013.6607464.

### Supplemental Information

Supplemental information for this article can be found online at http://dx.doi.org/10.7717/peerj-cs.1053#supplemental-information.

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
