# Peer review of "Data augmentation and deep neural networks for the classification of Pakistani racial speakers recognition"

_PeerJ Computer Science, doi:10.7717/peerj-cs.1053_

## Round 0.1 · original submission · Major Revisions

Dear Authors,

Since reviewers are showing a lot of concerns in your article. Therefore, please revise and resubmit for evaluation.

Reviewer 1 ·

Basic reporting

In this study, the authors present a data augmentation method that shifts the pitch, uses multiple window sizes, stretches the time, and adds white noise to the original audio.

The presented subject is undoubtedly attractive. Just the presented study might be somewhat weak in innovation. Frankly, the proposed approach might not be technically sound.

Experimental design

Transfer from previous research to deep neural networks, more like an MLP (multilayer perceptron), which stacks too many “Dropout” and “ReLU” layers, which might be somewhat unintelligible.

Data augmentation for fitting deep learning has been well-known efficacy most of the time.

Validity of the findings

Problem formation of this study might not be credible. Therefore, the authors are required to address the potential of the presented study based on speech emotion recognition.

Particularly, making necessary efforts to let the presented novelties much more recognizable by other scholars.

Additional comments

None

·

Basic reporting

The paper addresses an interesting problem of Speech Emotion Recognition. Leveraging Deep Neural Networks, it proposes a technique to classify multiple Urdu language accent speakers. While the evaluation and analysis of the proposal show promising results, the language and structure (elaborated in **Additional Comment**) of the paper makes it extremely hard to read thus hindering fluency and comprehension.

I really appreciate the authors' efforts to share the code with the research community. I believe this will help further research and reproducibility of the proposed technique. It would be more beneficial if the author share the dataset (**Augmented Dataset**) for end-to-end reproducibility of the results.

Experimental design

No comment

Validity of the findings

Although the authors reported the results of the analysis, a comparison with related work would really help the validation of the findings. Also, it would be interesting to validate the results in different multilingual settings with lower vs. high pitches of multiple speakers etc.

Additional comments

Language, Structure and Sign-posting:

The lack of structure, use of confusing language, and absence of sign-posting made this paper a bit harder to read. For instance, in the abstract, you first need to clearly define the problem and mention why it is important to solve it. You also need to sign-post your take on related work and the need for an improved, efficient mechanism to solve the problem. You then need to structure and briefly present your technique and salient features (i.e., some results). At the end you need to highlight the key takeaway(s) from your research.

The aforementioned structure could be followed throughout the (sub)sections of the paper.

From the abstract, it is not clear how and why SER is a complex issue? For clearer readability, perhaps, you may define SER before terming it as a complex issue. Once defined, you can then pin-point the complexity and attribute it to the main component(s) in SER.

What do you mean by limited database? Do you mean a limited number of speech emotion dataset?

You need to clarify why **seven**-layer framework was employed? Why not less or greater than **seven** a layer framework was used?

Introduction:
Please structure your introduction as per suggestion mentioned above.

Please consider restructuring the last paragraph of Introduction. For instance, "This report (--> paper) is divided .... about the conclusion of the research outcomes." could be a separate paragraph.

What do you mean by "numerous investigations"? Is numerous equals 10, 100, 1,000, etc?

Related Work:
- "Many countries ..." need to be correctly sign-posted. Countries do no research. Rather researchers investigate problems.

Having too many references without proper discussion or structure of the reference is concerning. Please avoid tangential references and cite the key, related work.

Data Augmentation
I think this section should be renamed as Data Manipulation as some techniques such as "Pitch Shifting" can not be termed as data augmentation technique. "Pitch Shifting" is a perturbation technique rather than data augmentation.
Please duplicate references e.g.,:

Please consider restructuring this section as the first paragraph is way too long and hard to read.

Methodology:

What does **0:42* represented in Table 2 (in Column, "Duration per sample")? Please keep consistency when reporting numbers or data in tables.

Salamon, J. and Bello, J. P. (2016). Deep convolutional neural networks and data augmentation for
environmental sound classification. CoRR, abs/1608.04363.

Salamon, J. and Bello, J. P. (2017a). Deep convolutional neural networks and data augmentation for environmental sound classification. IEEE Signal Processing Letters, 24(3):279–283.

Salamon, J. and Bello, J. P. (2017b). Deep convolutional neural networks and data augmentation for
environmental sound classification. IEEE Signal Processing Letters, 24(3):279283.

- And the following

Damskgg, E.-P. and Vlimki, V. (2017a). Audio time stretching using fuzzy classification of spectral bins. Applied Sciences, 7(12):1293.

Damskgg, E.-P. and Vlimki, V. (2017b). Audio time stretching using fuzzy classification of spectral bins. Applied Sciences, 7(12).


Misc:

It would be great if you could minimize the use of **passive voice**. In most of well-written, readable CS paper use **active voice** for improved communication.

Please align Table 7 with text-width size. You may reduce the width of column # 1 and column # 3.

---

## Round 0.2 · accepted · Accept

Based on the current quality of the article, I am happy to accept it for publication.